# Antifungal Properties of Two Volatile Organic Compounds on Barley Pathogens and Introduction to Their Mechanism of Action

**DOI:** 10.3390/ijerph16162866

**Published:** 2019-08-10

**Authors:** Amine Kaddes, Marie-Laure Fauconnier, Khaled Sassi, Bouzid Nasraoui, M. Haïssam Jijakli

**Affiliations:** 1Integrated and Urban Plant Pathology Laboratory, Gembloux Agro-Bio Tech (GxABT), University of Liège, 5030 Gembloux, Belgium; 2Laboratory of Chemistry of Natural Molecules, Gembloux Agro-Bio Tech (GxABT), University of Liège, 5030 Gembloux, Belgium; 3Department of Agronomy and Plant Biotechnology, National Agronomic Institute of Tunisia, University of Carthage, Tunis 1082, Tunisia; 4RL/Biogressors and Integrated Protection in Agriculture, National Agronomic Institute of Tunisia, University of Carthage, Tunis 1082, Tunisia

**Keywords:** methyl prop-2-enoate, methyl propanoate, biocontrol, volatile organic compound, mode of action

## Abstract

This study evaluated the antifungal effects of various volatile organic compounds (VOCs) against two common pathogens: *Fusarium culmorum* and *Cochliobolus sativus*. Among the various VOCs, methyl propanoate (MP) and methyl prop-2-enoate (MA) exhibited remarkable antifungal effects under different experimental conditions (direct or indirect contact) and at different concentrations (500–1000 μM). In addition, the type of antifungal effect (fungistatic or fungicidal) appeared to be strongly correlated with the VOC concentrations. Additional tests revealed that both molecules increased membrane permeability of pathogenic spores, which resulted in a decreased efflux of K^+^ ions into the intracellular medium.

## 1. Introduction

*Fusarium culmorum* and *Cochliobolus sativus* are known to cause root rot in different cereal varieties, especially wheat and barley. The incidence and severity of cereal contamination by these fungi have increased worldwide and resulted in serious yield losses. Damage caused by the contamination of sensitive barley cultivars by *Fusarium culmorum* and *Cochliobolus sativus* can raise up to 16–33% [1]. The damage induced by these two pathogens includes brown discoloration of the roots, coleoptiles, and sub-crown internodes of the host plants and mycotoxins contamination. Following the global ecologically sustainable approach to agricultural production, the use of chemical pesticides for cereal protection has grown more and more limited. Different eco-friendly techniques have been developed with a view to assessing alternative sustainable methods for cereal disease control [2,3,4,5]. In this context, many studies highlight the benefits of non-host crop techniques in managing foot and root rot [2,6,7]. This approach has attracted great attention and has been applied in different countries such as the United Kingdom and Australia [2,4,5,8,9]. At the same time, the development of biocontrol methods has also been suggested as an alternative method to fight against cereal pathogens. Globally, biocontrol methods use living organisms or natural substances produced by these organisms, such as pheromones and plant extracts, to inhibit pathogen growth [10]. Since the 1980s, volatile organic compounds (VOCs) produced by different endophytic fungi have attracted special interest due to their particular antifungal potential [11,12]. Fiers et al. [13] studied the effect of a spectrum of VOCs on the interaction between barley and the two main agents of root rot: *C. sativus* and *F. culmorum*. VOCs inhibited the growth of the pathogenic fungi. Interestingly, two VOCs, methyl prop-2-enoate (MA) and methyl propionate (MP), exhibited significant antifungal activities. These molecules inhibited fungal growth by more than 81% for both fungi [14]. Despite the unique ability of methyl prop-2-enoate (MA) and methyl propionate (MP) to inhibit a large spectrum of pathogens, their mechanisms of action still remains poorly studied. This prompted us to undertake the present study.

The main objectives of this work were (i) to evaluate the direct and indirect inhibitory effects of methyl prop-2-enoate (MA) and methyl propanoate (MP) on the growth of the two fungal strains *C. sativus* and *F. culmorum,* and (ii) to start deciphering the way these two molecules inhibit mycelial growth and spore germination.

## 2. Materials and Methods

### 2.1. Fungal Strains

*F. culmorum* (MUCL 28166) and *C. sativus* (MUCL 46854) strains were provided by the Belgian Co-ordinated Collection of Microorganisms (BCCM-MUCL) (Louvain-la-Neuve, Belgium). The strains were grown on PDA (Merck KGaA, Darmstadt, Germany) at 23 °C and were subjected to 16 hL:8 hD photoperiod.

### 2.2. Evaluation of the Effect of Five VOCs in Gas Phase on F. culmorum and C. sativus Growth Without Direct Contact

The antifungal activities of five VOCs—methyl propanoate, methyl prop-2-enoate, isobutylformate, p-cymene, and longifolene (Sigma-Aldrich)—were evaluated against the two pathogens *F. culmorum* and *C. sativus*. Different commercial solutions of VOCs (Sigma-Aldrich) were prepared to a final concentration of 500 µM. PDA (3.9%; *w*/*v*) and WA (1% agar (Difco, Grenoble, France); *w*/*v*) media were prepared and inoculated with 10-day-old cultures of *F. culmorum* and 3-week-old active *C. sativus* cultures. Each VOC was placed on a filter paper in a small 2 cm Petri dish with a lid in the middle. Thus, there was no direct contact between the pathogen and the VOCs, gases phase could be easily spread on the square-shaped petri dish (Greiner, Belgium). The petri dishes were placed in a growth chamber under LED light (94 mmol photons/m^2^/s) with a 16 hL:8 hD photoperiod for 10 days (Appendix A).

The radial growth (RG) of the fungi was determined by measuring two perpendicular diameters with a graduated ruler and averaging the values every 24 h until T = 240 h. A total of 15 Petri dishes was used for each VOC and each fungal strain. Each assay was independently replicated three times. Petri dishes were randomly placed in the culture chamber. The growth inhibition rate was calculated as follows [14]:Growth inhibition rate = (RG control − RG treated sample)/(RG control) × 100(1)

Statistical analyses were performed using Minitab 17 [15] (Minitab Inc., State College, PA, USA).

The growth rate of every fungus in the presence of each VOC was determined by an analysis of variance (*p* < 0.05) (AV1) using one-way ANOVA, followed by Tukey and Dunnett multiple comparison test (*p* < 0.05).

### 2.3. Evaluation of the Effect of Methylprop-2-Enoate and Methylpropanoate on F. culmorum and C. sativus Growth in Direct Contact

The effect of methylprop-2-enoate and methylpropanoate on *F. culmorum* and *C. sativus* growth in direct contact was determined according to methods described by Kaddes et al. [14], using PDA as the culture medium.

### 2.4. Evaluation of the Fungicidal/Fungistatic Effect of the Most Efficient VOCs on C. sativus and F. culmorum Growth

To evaluate the fungicidal/fungistatic effects of methylprop-2-enoate and methylpropanoate, both molecules were supplemented on PDA medium at two different concentrations (500 and 1000 µM) and poured in a 600 mL cell culture flask where a 70 mm disk of peripheric culture of each fungal strain had been placed. Three flasks were prepared for each concentration and were incubated at 23 °C and under a 16 hL:8 hD photoperiod for 10 days. The negative controls were VOC-free. After 10 days, when mycelium growth was not observed, the flask was opened, and the mycelial disk was transferred to another PDA Petri dish. All petri dishes were maintained at 23 °C and under the same photoperiod for 10 days to evaluate the effect of the VOCs. The experiment was performed in triplicate, and 3 flasks per compound and per concentration were analyzed in each replicate.

### 2.5. Evaluation of the Release of K^+^ Ions in the Extracellular Medium

The amount of K^+^ ions released in the extra-cellular medium was used as an indicator of the effect of MA and MP on the integrity of the fungal spore membranes. Five milliliters of spore suspensions at a concentration of 105 conidia/mL were prepared for each fungus in PDB medium. Methyl prop-2-enoate and methyl propanoate were added to obtain the following concentrations: 100, 500, and 1000 µM. A Quantofix^®^ kit of colored strips (Macherey-Nagel) was used to measure the release of K^+^ ions. The strips were used according to the protocol provided by the manufacturer (Appendix B). For each fungus, measurements were made every hour for 5 h.

### 2.6. Data Analysis

All experiments were performed in triplicate. Data were analyzed using analysis of variance (*p* < 0.05). Tukey and Dunnett multiple comparison test (*p* < 0.05) was performed using the Minitab 16.2.2 software [16].

## 3. Results

### 3.1. Evaluation of the Effect of Five VOCs on F. culmorum and C. sativus Growth

In the first part of the study, we evaluated the antifungal capacity of five VOCs (methyl propanoate, methyl prop-2-enoate, isobutylformate, p-cymene, and longifolene) against the two pathogenic strains *F. culmorum* and *C. sativus*. According to Fiers et al. [13], all five VOCs emitted *de novo* substances b infected barley roots.

The experiments were performed according to the direct contact protocol using WA as a culture medium. The VOC concentration (500 µM) was chosen on the basis of previous results reported by Kaddes et al. [14]. As can be seen in Figure 1A,B, the growth rate of both pathogens closely depended on the type of VOC added to the culture medium. Methylprop-2-enoate and methylpropanoate exhibited the highest antifungal potential against both pathogens. The growth rates of *F. culmorum* and *C. sativus* after application of methylprop-2-enoate for 240 h were estimated to be only 3% and 2%, respectively. The same behavior was also observed after application of methyl propanoate. In contrast, isobutylformate, longifolene, and p-cymene exhibited low antifungal activities against both fungal strains. Based on this first set of experiments, only methylprop-2-enoate and methyl propanoate were retained for further analyses.

### 3.2. Evaluation of the Inhibitory Effect of the Most Efficient VOCs on Fungal Growth in Direct and Indirect Contact

We evaluated the antifungal effect of methyl prop-2-enoate and methyl propanoate against *F. culmorum* and *C. sativus* in direct and indirect contact. Different experiments were performed on PDA medium or WA supplemented with 500 µM of each VOC before being inoculated with the appropriate fungal strain.

#### 3.2.1. Effect of Methyl Prop-2-Enoate and Methyl Propanoate in Direct Contact

Figure 2A,B presents the time course of the growth of the two fungal strains in PDA medium supplemented with VOCs or not. In the control medium, the growth diameter of the fungal strains increased and reached 1.24 cm and 12.54 cm for *F. culmorum* and *C. sativus* after 240 h of incubation, respectively. Despite this difference, supplementation of the culture medium with VOCs drastically inhibited mycelial growth. The growth rate of *F. culmorum* was estimated to be only 4% after application of methyl propionate or methyl prop-2-enotate. The percentage of inhibition of *C. sativus* mycelial growth ranged between 88.6% and 100% after application of methylpropionate and methylprop-2-enoate, respectively. Taking these results into account, we evaluated the antifungal potential of the two VOCs against *F. culmorum* and *C. sativus* in different experimental conditions (type of culture medium and type of contact).

A comparative study was conducted to evaluate the effects of the culture medium on the antifungal capacity of methylprop-2-enoate and prop-2-enotae. As shown by Figure 3A,B a comparison of *F. culmorum* and *C. sativus* mycelial growth in direct contact with the VOCs at 500 µM on water agar and PDA medium. The results of mycelial growth on WA medium were provided from the study by Kaddes et al. [14]. The results highlighted similar antifungal effects of methyl propanoate against *F. culmorum* on both culture media. Nevertheless, significant differences were observed with methylprop-2-enoate: *C. sativus* growth in its presence significantly depended on the culture medium. By contrast, no significant difference was observed under methyl prop-2-enoate treatment.

#### 3.2.2. Effect of Methyl Prop-2-Enoate and Methyl Propanoate in Indirect Contact

In the second part of the study, we evaluated the antifungal capacity of methyl prop-2-enoate and methyl propanoate on *C. sativus* and *F. culmorum* in indirect contact using PDA as a culture medium (Figure 4A,B). The results highlighted a strong inhibitory effect of both VOCs *against C. sativus* and *F. culmorum*. The percentages of mycelial growth inhibition after application of methyl propanoate reached 95% and 98% for *C. sativus* and *F. culmorum*, respectively. Methyl prop-2-enoate inhibited 99% of growth in both pathogens.

We then compared the indirect inhibitory effects of the two VOCs on PDA medium and WA medium. The antifungal effects of both molecules were completely independent of the media composition (Figure 5).

### 3.3. Effect of the VOC Concentrations on VOC Antifungal Ability

We conducted a new set of direct/indirect experiments to evaluate the effect of the VOC concentration of the antifungal capacity of methyl prop-2-enoate and methyl propanoate. The final concentration of each molecule was adjusted to 500–1000 µM in PDA/WA medium. Table 1 summarizes the results of the fungistatic/fungicide effect of each molecule at different concentrations. The mycelial growth of both pathogens was inhibited in the presence of methyl prop-2-enoate at 500 and 1000 µM. However, the mycelial growth of *C. sativus* was inhibited only in the presence of methyl propanoate at 1000 µM.

Implants [A1] that were treated with methyl prop-2-enoate and methyl propanoate at concentrations of 500 and 1000 µM during the first part of the test were transferred to PDA medium. In MP-treated cultures, mycelium of pathogens was observed in vials. This suggests that fungistatic effect of MP was weak. In contrast mycelial growth of both pathogens was completely inhibited when cultures were treated with MA. This suggests that MA exerts powerful fungicide effects.

### 3.4. Evaluation of the Release of K^+^ Ions into the Extracellular Medium

The above-mentioned results confirm the antifungal effects of methyl prop-2-enoate and methyl propanoate. Therefore, we tried to understand the mode of action of these molecules through a new set of experiments. The amount of K^+^ ions released into the extra-cellular medium was used as an indicator of the effect of MA and MP on the membrane of fungal spores.

Methyl prop-2-enoate and methyl propanoate had an effect on *F. culmorum* and *C. sativus* conidia that was related to the emission of K^+^ ions into the extracellular medium (Table 2 and Table 3, Appendix A).

After 3 h of incubation, the quantity of potassium released during treatment with methyl prop-2-enoate and methyl propanoate at 500 μM reached 700 mg/L. This quantity increased up to 1000 mg/L after 5 h of incubation with methyl prop-2-enoate at 1000 μM. Similar results were observed with *C. sativus* treated with either molecule.

## 4. Discussion

The first part of our study focused on the evaluation of the antifungal capacity of five VOCs, namely methyl propanoate, methyl prop-2-enoate, isobutylformate, p-cymene, and longifolene, on the two pathogenic fungi *F. culmorum* and *C. sativus*. Among the five VOCs, two organic esters (methyl prop-2-enoate and methyl propanoate) exhibited a particular antifungal potential in direct and indirect contact. In contrast, the antifungal effects of isobutylformate, p-cymene, and longifolene were negligible. These results are in accordance with those of Kaddes et al. [14], who reported that p-cymene exhibited a slight inhibitory effect when used in direct contact. This could be explained by its low molar mass, which makes it less volatile as compared to methyl prop-2-enoate and methyl propanoate. Moreover, our results confirm those of Kaddes et al. [14], who noted the highest percentages of pathogen growth inhibition under methyl prop-2-enoate and methyl propanoate treatment. According to these authors, methyl prop-2-enoate and methyl propanoate treatment at 500 μM resulted in up to 81% growth inhibition for *F. culmorum* and *C. sativus*, while p-cymene had a lower effect (73%) on both fungi [14].

In indirect contact, methyl prop-2-enoate had similar inhibitory effects on both fungal strains grown on PDA medium or WA medium. The same results were observed when the antifungal effect of methyl propanoate was evaluated against *C. sativus*. By contrast, the antifungal effect of methyl propanoate against *F. culmorum* was strongly dependent on the medium composition.

Furthermore, there was no significant difference between the inhibitory effect of methyl prop-2-enoate and methyl propanoate on *F. culmorum* in direct contact on PDA or WA. However, this was not the case for *C. sativus*, which developed better in the presence of methyl propanoate. This could be explained by the fact that *C. sativus* was resistant to methyl propanoate on PDA because environmental conditions were more favorable for its growth. This resistance could be related to the structure of its conidia, whose generally thick cell wall facilitates survival [17,18,19].

Methyl prop-2-enoate and methyl propanoate had a significant effect on *C. sativus*, especially on WA medium. This can be linked to the physiology of the fungus, as unfavorable environmental conditions can considerably affect *C. sativus* development. Neched [20] reported that the oxidative action of oxygen or humidified ozone generated oxidative stress and induced a clear decrease of the contamination level or even an absence of *C. sativus* on barley seeds. Poor environmental conditions therefore seem to make *C. sativus* more sensitive to VOCs.

Our results also show that methyl prop-2-enoate and methyl propanoate had a fungicidal or fungistatic effect against both pathogenic fungal strains in PDA medium. At a concentration of 500 µM, methyl prop-2-enoate had a fungicidal effect. A *Combretum racemosum* natural extract used in vitro at 2 g/L had a fungicidal effect on the mycelial growth of three telluric fungal pathogens of tomato crops in Ivory Coast(Zirihi et al. [21]). In comparison with that study, methyl prop-2-enoate had an important antifungal effect at lower concentrations (500–1000 µM).

The antifungal activity of MP and MA could be related to a cell disruption phenomenon. Both VOCs induced an increase in the membrane permeability of pathogenic spores and a decrease of the efflux of K^+^ ions into the intracellular medium [21]. To compensate for this imbalance, the activity of the proton pumps has to increase to ensure the efflux of H^+^ ions into the intracellular medium and maintain electrical charges on either side of the membrane at equilibrium [22,23,24]. This could induce a drastic change of the pH in the intracellular medium and inhibit fungal growth [18,19]. Many studies have highlighted the ability of nonanoic acid molecules and strobilurin to disturb the pH gradient between the intracellular medium and the extracellular media [25,26,27,28]. To our knowledge, this is the first study that has focused on the mode of action of these molecules on these two pathogens.

## 5. Conclusions

In the last few years, VOCs, a complex mixture of volatile compounds, have attracted great attention due to their antimicrobial potential. Their ability to be used as antifungal agents to fight against numerous pathogens has been considered as an attractive biocontrol strategy in agriculture. In the present study, we noticed the strong effect of two organic esters (methyl prop-2-enoate and methyl propanoate) against the two pathogens *Fusarium culmorum* and *Cochliobolus sativus*. We highlighted the ability of methyl prop-2-enoate to inhibit *F. culmorum* growth by more than 99% and 97% in direct contact phase and in gas phase, respectively. In the same way, methyl propanoate showed remarkable inhibitory activity against *F. culmorum*. As for *C. sativus*, its proliferation was totally inhibited by both organic esters in direct and indirect contact. In a second step, we attempted to understand the mechanism of action of methyl propanoate and methyl prop-2-enoate. After contact between fungal spores and the VOCs, an imbalance of the distribution of K^+^ ions between the intracellular medium and the extracellular medium was observed. This result suggests that both VOCs affect the integrity of the fungal membrane and disturb the pH gradient between the intracellular medium and the extracellular medium. However, further work is needed to confirm this hypothesis. In order to promote the use of methyl prop-2-enoate and methyl propanoate in agriculture, the development of cost-effective preparations is highly required. Alginate beads were recently used as vectors to ensure a slow diffusion of volatile molecules in the soil to fight against parasitoids and aphid predators [12,29]. The same strategy could be adopted to coat the antifungal molecules methyl prop-2-enoate and methyl propanoate and ensure their diffusion in the soil.

## Figures and Tables

**Figure 1 ijerph-16-02866-f001:**
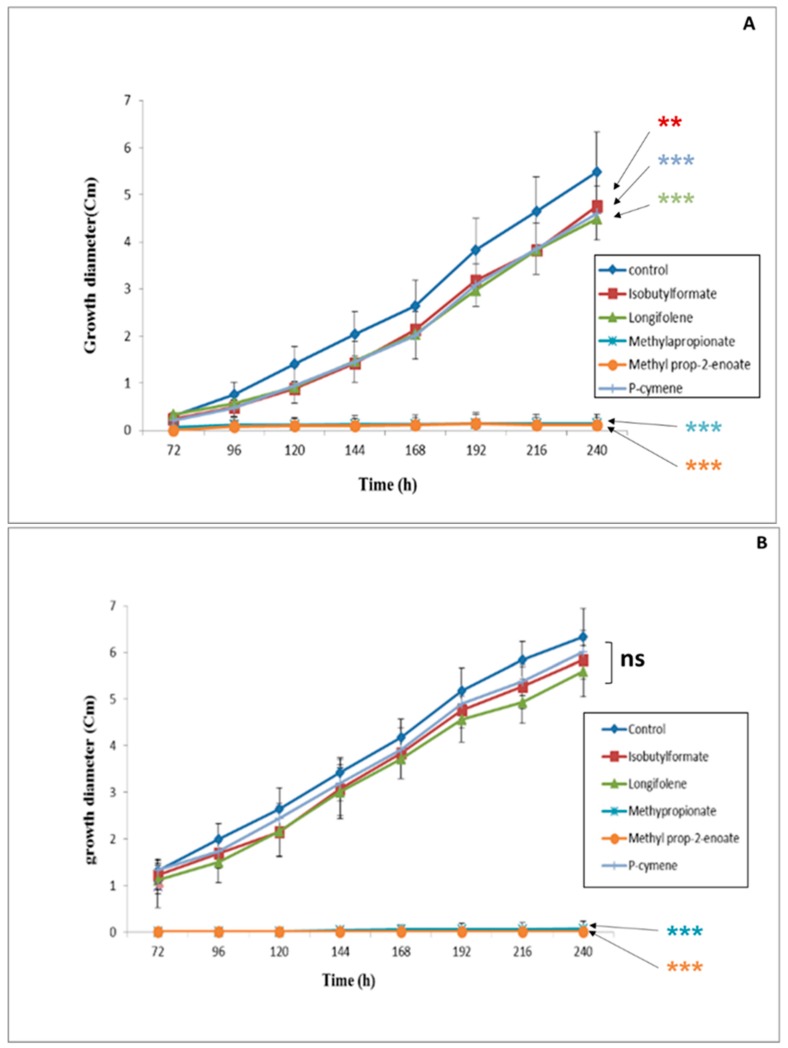
Growth of *F. culmorum* (**A**) and *C. sativus* (**B**) in the presence of volatile organic compounds (VOCs) at 500 µM in gas phase. (**): Significant result in comparison with the control, according to Dunnett’s test (*p* < 0.05). (***): Highly significant result in comparison with the control, according to Dunnett’s test (*p* < 0.05). ns: Non-significant result in comparison with the control, according to Dunnett’s test (*p* < 0.05).

**Figure 2 ijerph-16-02866-f002:**
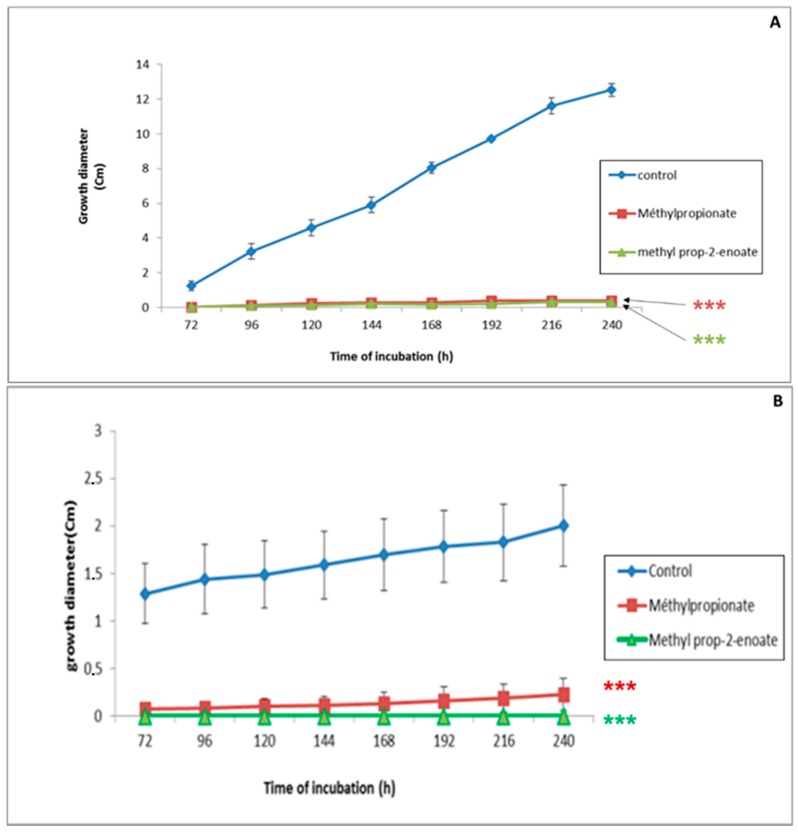
Evolution of *F. culmorum* (**A**) and *C. sativus* (**B**) mycelial growth in direct contact with methyl prop-2-enoate and methyl propanoate at 500 µM on PDA medium. (***): Highly significant result in comparison with the control, according to Dunnett’s test (*p* < 0.05).

**Figure 3 ijerph-16-02866-f003:**
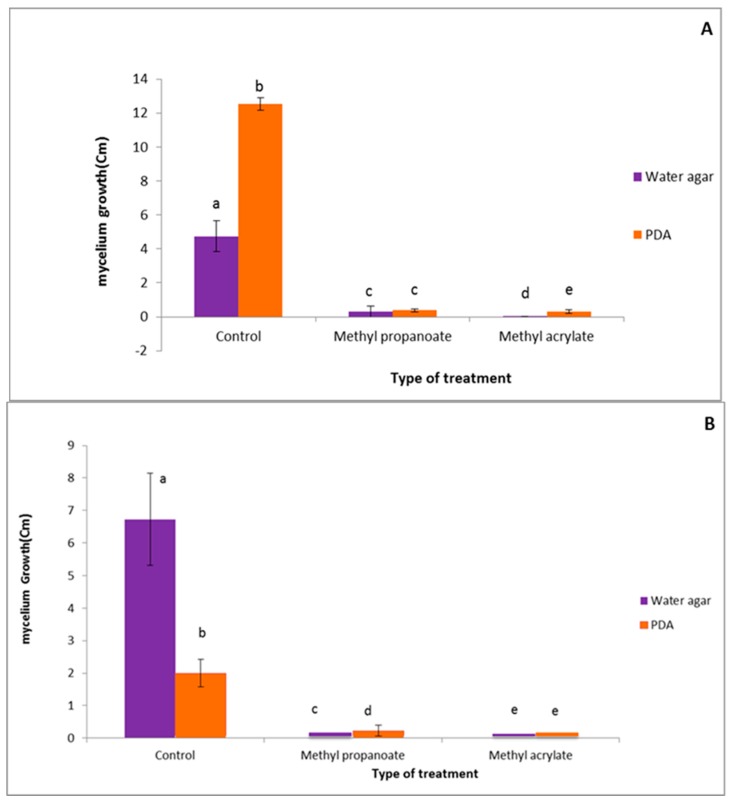
Comparison of *F. culmorum* (**A**) and *C. sativus* (**B**) mycelial growth in direct contact with the VOCs at 500 µM on water agar and PDA medium. Identical lower-case letters indicate non significantly different results according to Tukey’s test (*p* < 0.05).

**Figure 4 ijerph-16-02866-f004:**
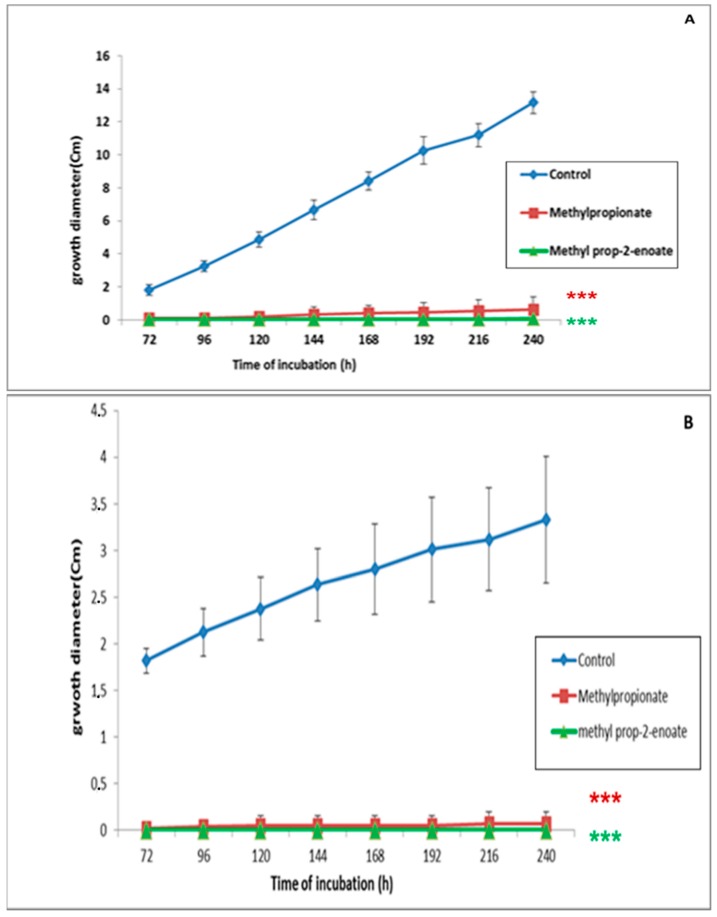
Evolution of *F. culmorum* (**A**) and *C. sativus* (**B**) mycelial growth in gas phase with MA and MP at 500 µM on PDA medium. (***): highly significant result in comparison with the control according to Dunnett’s test (*p* < 0.05).

**Figure 5 ijerph-16-02866-f005:**
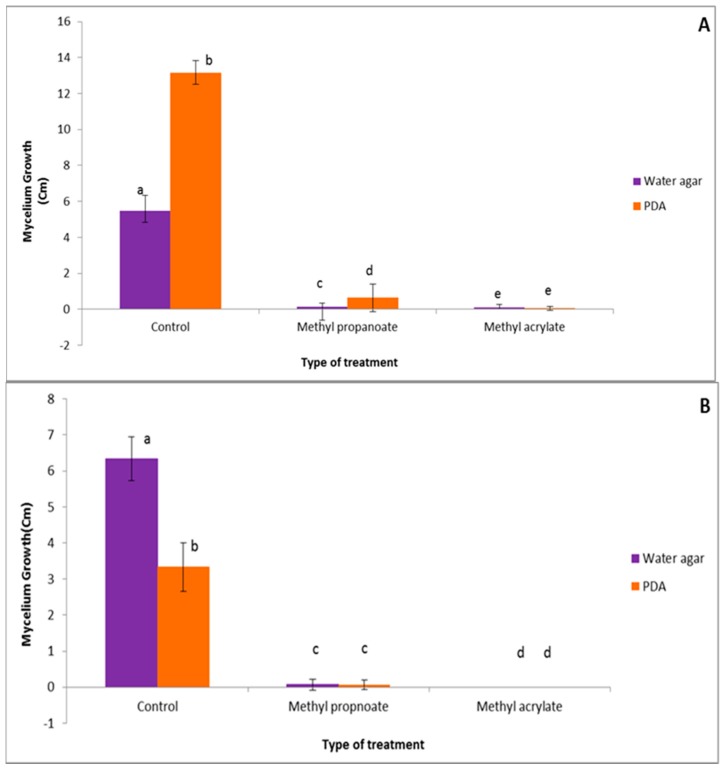
Comparison of *F. culmorum* (**A**) and *C. sativus* (**B**) mycelial growth in gas phase with VOCs at 500 µM on water agar medium and PDA medium. Identical lower-case letters indicate non significantly different results according to Tukey’s test (*p* < 0.05).

**Table 1 ijerph-16-02866-t001:** Fungicidal/fungistatic effect of MA and MP on *F. culmorum* and *C. sativus* at different concentrations. First part of the results (**A**), second part of the results (**B**).

**A**
**VOCs**	***Fusarium culmorum***	***Cochliobolus sativus***
MP	MA	MP	MA
**500 µM**	-	--	--	--
-	--	--	--
-	--	--	--
**1000 µM**	-	--	--	--
-	---	--	--
-	---	--	--
**Controls**	+	+	+	+
+	+	+	+
+	+	+	+
**B**
**VOCs**	***Fusarium culmorum***	***Cochliobolus sativus***
***MA***	***MP***	***MP***	***MA***
**500 µM**	-	+	+	-
-	+	+	-
-	+	+	-
**1000 µM**	-	+	+	-
-	+	+	-
-	+	+	-

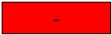
 Means “Absence of mycelium of the pathogen in vials”; 
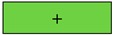
 Means “Presence of mycelium of the pathogen in vials”.

**Table 2 ijerph-16-02866-t002:** Amount of K^+^ ions (mg/L) present in the extracellular medium of a spore suspension of *Fusarium culmorum* subjected to different methyl prop-2-enoate and methyl propionate treatments for different incubation times.

Concentration	0 h	1 h	2 h	3 h	4 h	5 h
Control	400	400	400	400	400	400
MA 100 µM	400	400	400	400	400	400
MA 500 µM	400	400	400	700	700	700
MA 1000 µM	400	400	400	700	700	1000
MP 100 µM	400	400	400	400	400	400
MP 500 µM	400	400	400	400	400	700
MP 1000 µM	400	400	400	700	700	700

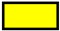
 400 mg/L K^+^ present in the extracellular medium of a spore suspension of *Fusarium culmorum*. 
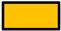
 700 mg/L K^+^ present in the extracellular medium of a spore suspension of *Fusarium culmorum*. 
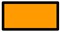
 1000 mg/L K^+^ present in the extracellular medium of a spore suspension of *Fusarium culmorum*.

**Table 3 ijerph-16-02866-t003:** Amount of K^+^ ions (mg/L) present in the extracellular medium of a spore suspension of *Cochliobolus sativus* subjected to different methyl prop-2-enoate and methyl propanoate treatments for different incubation times.

Concentration	0 h	1 h	2 h	3 h	4 h	5 h
Control	400	400	400	400	400	400
MA 100 µM	400	400	400	400	400	700
MA 500 µM	400	400	400	400	1000	1000
MA 1000 µM	400	400	400	400	1000	1000
MP 100 µM	400	400	400	400	400	700
MP 500 µM	400	400	400	400	700	1000
MP 1000 µM	400	400	400	400	700	1000

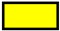
 400 mg/L K^+^ present in the extracellular medium of a spore suspension of *Cochliobolus sativus*. 
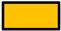
 700 mg/L K^+^ present in the extracellular medium of a spore suspension of *Cochliobolus sativus*. 
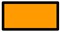
 1000 mg/L K^+^ present in the extracellular medium of a spore suspension of *Cochliobolus sativus*.

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
