# Peer review of "Antifungal Properties of Two Volatile Organic Compounds on Barley Pathogens and Introduction to Their Mechanism of Action"

_ijerph, 2019, doi:10.3390/ijerph16162866_

Round 1

Reviewer 1 Report

I am afraid the quality of the writing is so poor that I am unable to to carry out a proper review.  I am happy to review the paper if the authors have an English speaker correct it to a point where it is comprehensible.  

I believe the authors have good and interesting data, well worth publishing, although it is so poorly presented I cannot be entirely certain at this point.  For an example, take a look at table 1, where both colours shown apparently mean the same thing!  The other two tables are also confusing, and the large part of the discussion dedicated to explaining the K ion results is very wordy but does not actually state a conclusion.

The authors need to check their figure referencing, since in places the figures are referenced in the text in the wrong place.  They also need to improve their paragraphing in the introduction, since at present the manuscript contains many one sentence paragraphs.  This makes the introduction feel very disjointed.

I would like to know exactly how they carried out their controls, since VOCs are also emitted from media.

The size of the fungal discs used for the experiments seems very large (70 mm, presumably in diameter), that represents a large part of the space present in a 12 cm square Petri dish.  Is this correct?

Some of the conclusions presented do not make sense, but I very much feel that with the help of a native English speaker and careful reading on the part of the authors these issues can be resolved.

Author Response

1. Improve the writing style

As suggested by reviewer 2, the paper was edited for English spelling and grammar by an English native speaker. Please found in attached file, the editing certificate from Ediatge.

2. The authors need to check their figure referencing

As suggested by reviewer 2, the referencing of figures was checked

3. They also need to improve their paragraphing in the introduction, since at present the manuscript contains many one sentence paragraphs.  This makes the introduction feel very disjointed.

As suggested by reviewer 2, the introduction section was revised.

Please refer to lines 30-59 in the manuscript.

4. I would like to know exactly how they carried out their controls, since VOCs are also emitted from media.

It has been shown that gelose medium do not induce the secretion of VOCs molecules. this result was already confirmed by kaddes et al. 2016 and refers et  Fiers al 2013. Moreover, similar results were obtained using PDA medium suggesting that this medium do not induce the secretion VOCs molecules also.

The size of the fungal discs used for the experiments seems very large (70 mm, presumably in diameter), that represents a large part of the space present in a 12 cm square Petri dish. Is this correct?

Please check figures S1, S2 and S3 in the supplementary file;

Reviewer 2 Report

I added comments in the text of manuscript

Title

Ok, please think about mechanism? 

Abstract

Ok, correct

Key words

Please correct together or separately in compounds?

Introduction

Well written,

You can mention other species of plants and pathogens:

Straw and stubble barley favors infestation by Fusarium spp., and limit biodiversity.

For example:

1.      GleÅ„-Karolczyk K., BoligÅ‚owa E., Antoniewicz J. 2018. Organic fertilization shapes the biodiversity of fungal communities associated with potato dry rot. Applied Soil Ecology, 129, 43-51. DOI: https://doi.org/10.1016/j.apsoil.2018.04.012  

Materials and methods

Line 109

Statistical analyses were performed using Minitab 17 – Please explain statistical?

Maybe three repetitions or three experiments ? Please explain  

Line 111

Please give aim of analysis?

Results

General Comment

Well written and lack information about aim of research, please supplement  

Discussion

Ok, correct

In future I proposed used cotoxicology test fir VOCs and compare to plants - effect

Please give figure on mechanism …

Conclusions

Ok, correct and please supplement information on proton pumps – You no analysis mechanism of proton pumps

Author Response

1. Title

Ok., please think about mechanism ? 

As suggested by reviewer 1, the title was  changed

Please refer to line 1  in the revised manuscript

2. Key words

Please correct together or separately in compounds  ?

As suggested by reviewer 1, keywords were separated

Please refer to line 26 in the revised manuscript

3. You can mention other species of plants and pathogens:

Straw and stubble barley favors infestation by Fusarium spp., and limit biodiversity.

For example:

1.      GleÅ„-Karolczyk K., BoligÅ‚owa E., Antoniewicz J. 2018. Organic fertilization shapes the biodiversity of fungal communities associated with potato dry rot. Applied Soil Ecology, 129, 43-51. DOI: https://doi.org/10.1016/j.apsoil.2018.04.012  

As thiswork focuses on two specific barley pathogens, we limited our literature survey to related informations on this purpose:

4. Line 109

Statistical analyses were performed using Minitab 17 – Please explain statistical ?

As suggested by reviewer 1, the statistical test was explained in the text

Please refer to line 85-87 in the revised manuscript

5. Line 111

Please give aim of analysis ?

As suggested by reviewer 1, the aim of the analysis was explained in the text

Please refer to line 108-109 in  the revised manuscript

6. Remarks mentioned in lines 298, 304-310 and 320 concerning the mechanism action of both VOCs molecules.

In the revised version we proposed an obvious explication for the mechanism of action of both molecules. Please refer to lines 279-286 in the discussion part and lines 300-303 in the conclusion. In order to confirm these observations, additional experiments (proton pumps,..) need to be conducted in the future.